# The Role of the Endometrium in Implantation: A Modern View

**DOI:** 10.3390/ijms25179746

**Published:** 2024-09-09

**Authors:** Pavel I. Deryabin, Aleksandra V. Borodkina

**Affiliations:** Mechanisms of Cellular Senescence Laboratory, Institute of Cytology of the Russian Academy of Sciences, Tikhoretsky Ave. 4, Saint-Petersburg 194064, Russia

**Keywords:** embryo sensing, endometrium, spontaneous decidualization, implantation, decidual cells, personalized embryo transfer, WIN test, ERA

## Abstract

According to the current data, the endometrium acts as a “sensor” of embryo quality, which promotes the implantation of euploid embryos and prevents the implantation and/or subsequent development of genetically abnormal embryos. The present review addresses the nature of the “sensory function” of the endometrium and highlights the necessity for assessing its functional status. The first section examines the evolutionary origin of the “sensory” ability of the endometrium as a consequence of spontaneous decidualization that occurred in placental animals. The second section details the mechanisms for implementing this function at the cellular level. In particular, the recent findings of the appearance of different cell subpopulations during decidualization are described, and their role in implantation is discussed. The pathological consequences of an imbalance among these subpopulations are also discussed. Finally, the third section summarizes information on currently available clinical tools to assess endometrial functional status. The advantages and disadvantages of the approaches are emphasized, and possible options for developing more advanced technologies for assessing the “sensory” function of the endometrium are proposed.

## 1. Introduction

Assisted reproductive technologies (ARTs), particularly in vitro fertilization (IVF), have become established in a clinical practice as the most effective method for treating infertility. Despite significant advancements in IVF since the advent of this technology in 1978, to date, success rates have not surpassed 30–40% [1]. The success of ART programs primarily depends on the quality of oocytes, the quality of sperm, and the quality of the resulting embryos [2]. The utilization of preimplantation genetic testing to select euploid embryos has led to an increase in pregnancy rates [3]. Nevertheless, the implantation failure rates still not exceed 50–70% levels [3]. Recent findings suggest that one of the crucial yet often overlooked causes of implantation failure is an impaired functioning of the endometrium [4,5,6]. Traditionally, the endometrium was viewed as an effector tissue, which is transformed into a passive substrate for trophoblast invasion under the exposure of steroid hormones produced by the ovaries [7,8]. Current data suggest that the endometrium is capable of reacting differently to embryonic signals and acts as a sensor of embryo quality [8,9,10,11]. On the one hand, it creates favorable conditions for the implantation of euploid embryos; on the other, it prevents the implantation and/or subsequent development of genetically unstable embryos. About 40–60% of embryos are estimated to fail to implant as a result of aneuploidy [12]. These findings highlight the importance of a high-quality selection of embryos mediated by the maternal endometrium. This review reveals the evolution of the “sensory” ability of the endometrium, examines the implementation of this function at the cellular level, and summarizes the existing clinical applications of this fundamental knowledge to improve the efficiency of implantation in ART cycles.

## 2. “Spontaneous” Decidualization of the Endometrium Is an Evolutionary Sign of the Transition of Control over Implantation from the Embryonic to Maternal One

Even though mammalian viviparity evolved in the stem lineage of therian mammals, marsupials and eutherians employ distinct reproductive strategies, with a key difference in the establishment of the fetal–maternal interface [13,14,15,16] (Figure 1). For most of the duration of pregnancy in marsupials, the embryo is enclosed within an eggshell, while the contact between the fetal membranes and the uterine luminal epithelium occurs towards the end of the pregnancy, quickly followed by parturition. In most marsupials, the embryonic portion of the placenta is derived from the yolk sac, which may come into direct contact with the maternal endometrium but does not invade it. In contrast, embryo attachment during eutherian pregnancy leads to the establishment of a stable fetal–maternal interface, which is essential for the formation of a definitive chorioallantoic placenta capable of sustaining prolonged pregnancy [13,14]. Intriguingly, in both marsupials and eutherians, fetal–maternal attachment is associated with an inflammatory process; however, in marsupials, this results in parturition, whereas in eutherians, it leads to the establishment of a sustained fetal–maternal interface [15,16]. Current data suggest that this evolutionary transformation originates from a novel eutherian cell type—decidual stromal cells—which mediate the suppression of the ancestral inflammatory response and enable the endometrium to maintain the embryo for a prolonged period. For instance, human decidual stromal cells have been recently shown to suppress IL17A production by T cells, thereby preventing infiltration of neutrophils into the tissue [15]. Additional evidence regarding maternal adaptations to suppress uterine inflammation is described in the outstanding review [16]. Therefore, decidualization is a distinct feature of eutherian pregnancy, absent in marsupials (Figure 1).

The intensity of decidualization of the endometrium is associated with the depth of embryo invasion into the tissue [17,18]. Depending on the degree of endometrial invasion, three main types of eutherian placenta are recognized: hemochorial (the most invasive type, where trophoblast cells penetrate the endometrial stroma and its vasculature to access maternal blood), endotheliochorial (a less invasive type, where the trophoblast interfaces with endometrial endothelial cells), and epitheliochorial (a non-invasive type, where the trophoblast interfaces with luminal epithelial cells of the endometrium) [19]. Expectedly, in cetaceans and ungulates with a non-invasive type of placenta, decidual cells are not detected [20]. In contrast, in carnivores with an endotheliochorial type of placenta, small, isolated groups of decidual cells are found [18]. The most pronounced decidual reaction is observed in animals with a hemochorial type of placenta [18]. Notably, in the vast majority of animals characterized by decidualization, this reaction develops directly at the site of embryo implantation in response to the interaction between trophoblast cells and endometrial epithelial cells [16,21]. In other words, the decidual response of the maternal endometrium is controlled by signals emanating from a blastocyst.

A different picture is typical among menstruating animals, including humans, apes, elephant shrew, and certain bats [22,23,24]. For these species, decidualization is termed “spontaneous” because it occurs independently of contact with an embryo and is induced by maternal endocrine stimulation, specifically through the steroid hormone progesterone (P). According to recent findings, the responsiveness of endometrial stromal cells to P has evolved due to the continuous donation of the progesterone receptor (PR) binding sites to the regulatory regions of decidual genes by transposable elements in the eutherian lineage [25,26]. In humans, “spontaneous” decidualization begins during the luteal phase of the menstrual cycle, within the first few days following ovulation, and is driven by P produced by the corpus luteum appearing at the site of the ruptured dominant follicle [16,27]. In a conception cycle, the implanting embryo secretes a significant amount of human chorionic gonadotrophin (hGC), an embryonic anti-luteolytic signal that sustains ovarian P production until the definitive placenta is formed. In non-conception cycle, the absence of embryonic hGC leads to the demise of the corpus luteum and a subsequent drop in the P levels, resulting in menstruation [16,27]. The initial phase of menstruation occurs in decidualized endometrial stromal cells, which sense P withdrawal due to PR expression [27]. Decidual stromal cells initiate a highly regulated inflammatory response that results in the production of various inflammatory cytokines, chemokines, and pro-inflammatory prostaglandins [28]. These inflammatory events recruit and activate immune cells, leading to extracellular matrix degradation and piecemeal tissue breakdown [29]. Therefore, the development of “spontaneous” decidualization is thought to mediate the emergence of a novel evolutionary feature—menstruation—the rejection of an unclaimed part of decidualized endometrial functionalis layer in response to a drop in P levels during cycles without conception [16]. Such an energy-consuming mechanism has evolved in only a few animal species during evolution and is believed to provide them with certain advantages.

According to modern concepts, the emergence of “spontaneous” decidualization has mediated the shift of the control of the decidual reaction from the embryonic to the maternal one and thus underlined the development of the “sensory” ability of the maternal endometrium to an embryo [17]. This point of view finds a number of experimental confirmations. For instance, in vitro models of implantation revealed that decidualizing human endometrial stromal cells (EnSC) exhibit a robust response to damaged embryos [23,24]. Specifically, the composition of factors secreted by decidual cells changes significantly in the presence of genetically abnormal embryos [24]. The authors of this study conclude that “spontaneous” decidualization evolved to counterbalance the high incidence of chromosomal abnormalities in human embryos and enabled the maternal organism to restrict the implantation of “low-quality” embryos. This hypothesis was further supported by the findings indicating that the endometrium of women with an impaired decidual reaction exhibits reduced sensory function, which leads to increased implantation rates but recurrent miscarriages [24,30,31]. Indeed, recurrent pregnancy loss has been associated with stem cell deficiency, accelerated stromal senescence, and deficient decidualization that limit the differentiating capacity of the endometrium and predispose individuals to pregnancy failure [31]. Additional evidence links defective functioning of EnSC and impaired decidualization to various complications during pregnancy [32,33,34]. For example, both endometriosis and adenomyosis have been noted to express aggressive endometrial stem cells that display greater invasiveness [33]. Another pathology associated with defective decidualization is placenta accreta, a condition characterized by excessive invasion of the placenta into or through the uterine wall [34]. Since decidualization serves as a defense mechanism that protects the maternal organism from excessive embryo invasion, defects in this process can permit excessive placental invasion [32].

Thus, the presented data demonstrate that the acquisition of the trait of “spontaneous” decidualization during evolution enabled the human endometrium to develop sensory abilities for selecting “high-quality” embryos (Figure 1). The following section will delve into cellular mechanisms underlying the sensory ability of the endometrium.

## 3. Implementation of the “Sensory” Function of the Endometrium at the Cellular Level

The endometrium is the inner mucous membrane lining the uterus. At the histological level, it consists of a layer of columnar epithelium and supporting stroma [35,36]. During implantation, the embryo penetrates the epithelial barrier and then invades the underlying stroma [35,36]. Traditionally, the initial penetration stage is thought to mediate success of implantation, while the role of correct decidualization of the stromal compartment is often underestimated [17]. This idea is based on the results of in vivo studies performed on mice and rats [37,38,39]. Indeed, in the vast majority of rodents, only the penetration of the embryo through the epithelial barrier triggers decidual transformation of the tissue [37,38,39]. However, with the emergence of “spontaneous” decidualization, the sequence of these events has inverted. In humans, decidualization anticipates implantation and creates conditions for subsequent attachment and invasion of an embryo in the conception cycle [11,16,17]. In other words, the selective ability of the human endometrium is mediated by cyclic decidualization of the underlying stroma.

The main structural and functional component of the endometrial stroma is EnSC. EnSC differentiate into decidual cells in response to rising P levels following ovulation. Initially, decidualization was considered a unidirectional hormone-regulated process [40]. However, sequencing of single cells/nuclei analyses revealed that differentiation of EnSC results in the formation of multiple subpopulations of decidual cells [41,42]. Moreover, the delicate balance among these subpopulations mediates the receptivity/selectivity of endometrial tissue to an implanting embryo [41,43]. A brief introduction in the timing of EnSC decidual transformation is presented in Figure 2.

At the initial stage of decidualization, so-called pre-decidual cells are formed in response to increasing levels of P and intracellular cyclic adenosine monophosphate [17]. It is assumed that pre-decidual cells encapsulate the embryo through active migration and thus ensure its immersion into the thickness of the tissue [17,18]. Along the differentiation progression, most pre-decidual cells enter the mature state and form a major subpopulation of P-dependent decidual cells that express prolactin and IGFBP1 [17]. During the conception cycle, it is this type of decidual cell that provides nutrition and immunological protection to the implanting embryo and later forms the maternal decidual component of the placenta [17,41].

Following the appearance of mature decidual cells in the endometrium, a minor subpopulation of senescent decidual cells emerges [41,43]. Cellular senescence is a physiological response of cells to a stressor [44,45]. Regardless of the inducer, the activation of the senescence program in cells does not affect their viability; however, it results in an irreversible loss of proliferation and significant changes in the repertoire of secreted factors [44,45]. The latter is called the senescence-associated secretory phenotype and predominantly consists of pro-inflammatory factors and factors that promote extracellular matrix degradation [45]. The effects of senescence on individual tissues depend on the duration of the presence of senescent cells in a tissue and the efficiency of their clearance by immune cells. Therefore, an acute senescence mediates tissue remodeling in normal physiological processes such as embryonic development, placenta formation, and wound healing [45]. The chronic senescent cell presence promotes inflammation, mediates the progression of age-associated pathologies, and accompanies the aging of the organism [45].

According to the current perspective, senescent cells appear during decidualization as a result of a stress response to intense hormonal stimulation of a minor part of the EnSC population [41,42]. Due to the development of the senescence-associated secretory phenotype, the emergence of senescent decidual cells represents a crucial element in the preimplantation remodeling of endometrial tissue [17,43]. The dynamics of the presence and effects of senescent decidual cells are mediated by their close cooperation with mature decidual cells and immune cells. Notably, the enrichment of the endometrial stroma with a subpopulation of senescent decidual cells in a normal conception cycle is temporary [46]. This emergence peaks at the middle of the secretory phase and corresponds to the “implantation window”. However, towards the end of the secretory phase, the proportion of these cells decreases due to their elimination by uterine natural killer cells, which presence in the endometrium peaks in the late secretory phase [41,46].

Experimental findings suggest that the timely appearance of the above-mentioned subpopulations and the correct ratio among them are critical for embryo implantation. The first stage of the selection occurs before an embryo is embedded in the endometrial tissue, when motile pre-decidual cells surround the embryo and attach to its polarized trophectoderm cells [30,47,48,49]. “High-quality” embryos were shown to stimulate migration of pre-decidual cells, while “low-quality” embryos were not [30,47,48]. Conversely, the migration of undifferentiated EnSC is suppressed by signals emanating from “high-quality” embryos but not from “low-quality” embryos [30,49]. These data indicate that the selective ability of the endometrium to support the implantation of genetically correct embryos develops during decidualization. Further transition of pre-decidual cells into mature decidual cells results in a loss of migratory activity and the ability to attach to the embryo [17]. The next stage of the selection coincides with the appearance of senescent decidual cells. Due to secretory activity, these cells reorganize the extracellular matrix and create a conducive environment for the growth of the embryo and its interaction with mature decidual cells [43]. Deficiency of senescent decidual cells accelerates the transition of pre-decidual cells into mature ones and results in the entrapment and collapse of blastocysts in an excessively rigid matrix produced by mature decidual cells [43]. Such an imbalance is thought to contribute to recurrent implantation failures. In turn, an excess of senescent decidual cells facilitates embryo invasion due to the secretion of proteases and disintegration of the stromal matrix [43]. However, subsequently, such a loose matrix peels off, contributing to miscarriage, similar to what happens in cycles without conception during menstruation [43,50].

The subsequent fate and functional role of senescent decidual cells differ in cycles with and without conception. In cases of successful fertilization and implantation, senescent cells are eliminated from the endometrium by uterine natural killer cells [17,43]. In cycles without conception, pro-inflammatory senescent cells induce secondary senescence in neighboring mature decidual cells. As a result, almost the entire functional layer of the tissue becomes senescent, which in turn contributes to its loosening and shedding during menstruation [16].

In light of the described data, the importance of the endometrium in implantation becomes evident and highlights the necessity in clinical approaches to assess a functional status of the endometrium in an infertility treatment.

## 4. The Issue of Assessing the Functional Status of the Endometrium

The endometrium becomes receptive for the implanting embryo for a short period of time called the “implantation window” [51,52]. According to one of the earliest ART studies dated to 1992, transferring embryos during a period of 4 days from about 6 days after the luteinizing hormone (LH) peak results in a pregnancy with a higher probability than transferring embryos earlier or later than this period [51]. Current studies reveal even shorter duration of the period, approximately 30–36 hours between LH  +  6 and LH  +  9 in the natural cycles or between P  +  4 and P  +  7 in the hormonal replacement therapy cycles [52].

The traditional methods used to assess the functional state of the endometrium include ultrasound imaging of the pelvic organs and pipelle biopsy of the tissue [53,54]. These methods examine structure and thickness of the endometrium, as well as specific molecular markers, primarily the expression of steroid hormone receptors, using immunohistochemical staining. Both approaches are highly operator-dependent and are therefore susceptible to significant biases arising from intra- and interobserver variations. To standardize interpretation and to enhance the reproducibility of these characteristics, more sophisticated approaches based on artificial intelligence (AI) are being developed [55,56,57,58,59]. For example, a recent study described a novel tool called EndoClassify, which utilizes AI to analyze ultrasonographic imaging to select optimal endometrial development prior to embryo transfer [56]. Another study exploits ultrasound images of the endometrium to train and test several AI models, with implantation as a binary outcome [57]. Furthermore, AI models are applied to identify CD138+ plasma cells within endometrial tissue and to assess endometrial histology features by calculating the areas occupied by epithelial and stromal cells [58,59]. Therefore, AI has the potential to improve the objectivity of ultrasonographic and histological examinations of the endometrium for both research and clinical purposes. At the same time, to date, there is no compelling evidence of a significant correlation between the changes observed at the morphological/histological levels of the endometrium and the success of implantation in either natural or stimulated cycles [53,54,60].

The rise of omics technologies has opened new prospects for assessing endometrial receptivity [52,61]. To date, seven instruments fulfilling this task are commercially available on the market [61]. However, only four instruments have data published in a peer-reviewed journal, and basic information about these tools is summarized in Table 1 [62,63,64,65]. Generally, the tools rely on reference endometrial gene expression profiles and a selected panel of marker genes whose expression corresponds to the “implantation window” for a specific patient. For almost all instruments, the biopsy procedure is performed on the seventh day after the LH peak in the natural cycles or on the fifth day after the beginning of P intake in the stimulated cycles. The samples are placed in a preservation buffer and transported to a laboratory for analysis, which takes from 5 days to 3 weeks depending on the technology used to assess gene expression. Based on the results of the analysis, a conclusion is drawn about whether the endometrium corresponds to one of several conditions, usually including receptive and non-receptive states (Table 1).

## 5. Discussion and Future Perspectives

As can be seen from Table 1, the first tests for the assessment of the endometrial receptive state were introduced more than a decade ago. Nevertheless, the rationale towards the application of these instruments is still debatable. In original studies, most tests demonstrated promising results, increasing the chances of clinical pregnancy and childbirth in patients attending ART centers [62,63,64,65]. Meanwhile, further independent studies reported less consistent results in the application of the tools (summarized in [52,70]). Notably, a clear lack of prospective and randomized controlled trials to validate the effectiveness for the majority of tests was observed. Although the idea of the personalized embryo transfer timing estimation is generally espoused, the implementation of the developed tests harbor criticism of a community [52,70]. We believe that the effectiveness and wider adoption of these tools could be limited by two major interconnected factors stemming from the design of these tools.

Firstly, these tests have not been adjusted for the variability in the length of the menstrual cycle between donors when being built based on the input training data. Secondly, standard endometrial biopsy is a mixture of luminal and glandular epithelium, stromal cells, and blood cells, and the gene signatures identified for endometrial receptivity in these tests fail to consider the heterogeneity of the cellular composition of the analyzed samples and do not represent specific cellular and systemic processes. Thus, according to the recent analysis, the gene lists identified in different studies as the markers of endometrial receptivity have a minimal overlap [61]. This discrepancy may be attributed to the difference and small sizes of the training data used to create the predictors and to the heterogeneity of the cellular composition of the endometrial samples.

In light of these limitations, it seems rational to focus on several ideological and technical points. Firstly, two algorithms were recently developed to estimate the relative progression of the menstrual cycle among different donors based on modeling of the transcriptional changes in a bulk endometrial data [71,72]. The use of these algorithms may allow unification of the input transcriptomic profiles used to determine gene signatures. Secondly, analysis of single cells/nuclei sequencing data from endometrial samples can provide a clearer picture of the expression of specific genes in specific cell types [73]. And lastly, as described in the previous two chapters of this review, correct stromal decidualization plays a key role in the success of implantation and thus can be used as a relevant and specific cellular process reflecting transition of the endometrium into the receptive state. In this regard, when developing a gene signature of endometrial receptivity, special attention should be paid to the expression profile of decidualizing stromal cells. Taking into account the above points may contribute to the improvement of current approaches or the creation of completely new technologies for assessing endometrial receptivity with more precision and interpretability, thus increasing the effectiveness of ART cycles.

## Figures and Tables

**Figure 1 ijms-25-09746-f001:**
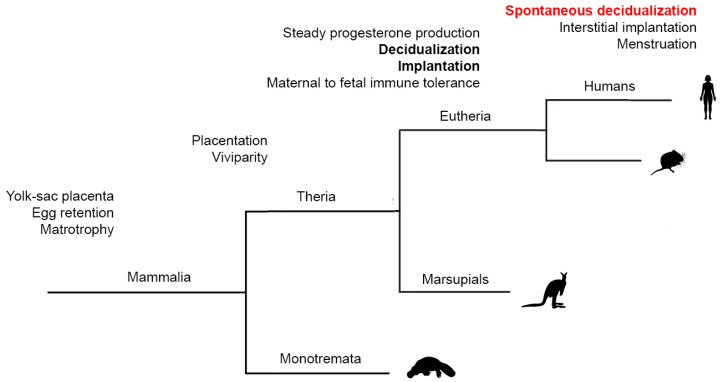
Schematic dendrogram reflecting the emergence of spontaneous decidualization during the evolution of mammals (adapted from [16]).

**Figure 2 ijms-25-09746-f002:**
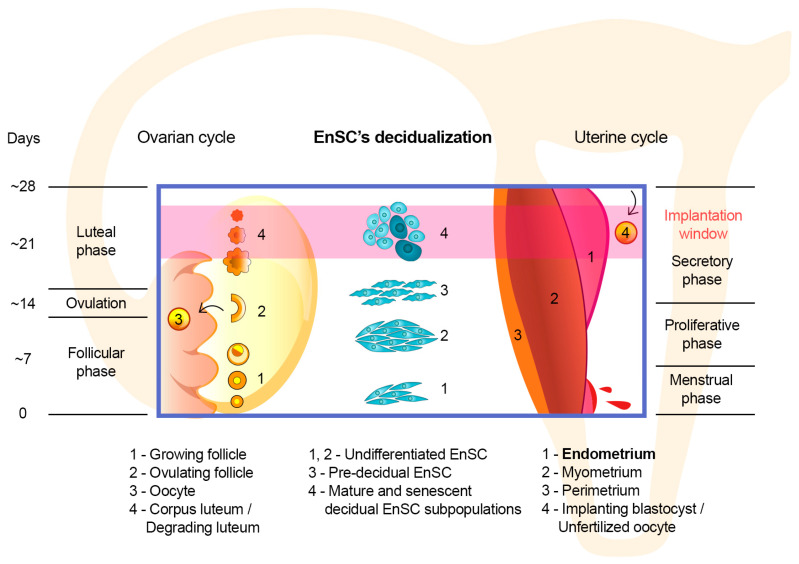
Dynamics of decidualization of endometrial stromal cells (EnSC) relative to the progression of the menstrual and ovarian cycles in humans.

**Table 1 ijms-25-09746-t001:** Commercially available tests for the assessment of the receptive state of endometrial tissue.

Name, Year	Number of Genes, Method	Result	Evidence of the Efficiency
WIN-test, 2009 [62]	11 genes, qPCR	3 categories: receptive, partially receptive, non-receptive tissue	Two retrospective and one prospective trial [66,67,68]. Key findings: an increase in clinical pregnancy, ongoing pregnancy, and live birth rates when applied to IVF patients with recurrent implantation failures.
ERA, 2011 [63]	238 genes, microarray	6 categories: proliferative, early-receptive, partially receptive, receptive, late-receptive, or post-receptive tissue	Thirteen retrospective, two prospective, and one randomised controlled trial (summarized in [61]). Key findings: an increase in clinical pregnancy and live birth rates when applied to IVF patients with recurrent implantation failures; no significant shifts in the general population.
ER Map/ER Grade, 2018 [64]	40 genes, qPCR	5 categories: proliferative, pre-receptive, receptive, late-receptive, or post-receptive tissue	One retrospective trial [69]. Key findings: the probability of clinical pregnancy during embryo transfer at the predicted receptive state of the endometrium is higher than at the moment that has been estimated as non-receptive state in the general population.
beReady, 2019 [65]	67 genes, targeted allele counting by sequencing	4 categories: pre-receptive, early-receptive, late-receptive, or post-receptive tissue	No clinical trials. According to the original study, the test is effective in predicting the shift of the “implantation window” when applied for IVF patients with recurrent implantation failures [65].

## Data Availability

No new data were created or analyzed in this study. Data sharing is not applicable to this article.

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
