# Peer review of "The Role of the Endometrium in Implantation: A Modern View"

_ijms, 2024, doi:10.3390/ijms25179746_

Round 1

Reviewer 1 Report

Comments and Suggestions for Authors

The authors clearly and objectively addressed the importance of the endometrium in the implantation process. The work is well written and well-founded, therefore, we are in favor of its acceptance.

Author Response

COMMENT: The authors clearly and objectively addressed the importance of the endometrium in the implantation process. The work is well written and well-founded, therefore, we are in favor of its acceptance.

RESPONSE: We would like to thank the respected Reviewer for the appreciation of our study.

Reviewer 2 Report

Comments and Suggestions for Authors

The manuscript presents an intriguing idea, yet the content would benefit from a more detailed exploration of the underlying mechanisms, moving beyond general descriptions to focus on specific biological processes. 

Lines 45-57: While it is widely recognized that decidualization varies across species, the manuscript should elaborate on these differences with greater specificity. For instance, it would be valuable to discuss variations in trophoblast types or unique gene manipulations that occur during this stage. Additionally, the manuscript should explore the cellular interactions involved and the signaling pathways through which the blastocyst communicates with the maternal endometrium. A concentrated figure or table summarizing these processes would enhance the reader's understanding. 

Lines 58-59: The relationship between hormonal regulation, menstruation, and decidualization should have a more detailed update. The manuscript needs to include recent findings from current studies, providing a comprehensive overview of how these factors interplay during the decidualization process. 

Lines 71-84: The manuscript should delve deeper into the genetic changes that drive "spontaneous" decidualization, specifying the extent of these alterations. Additionally, the discussion around recurrent miscarriages and their role in restricting the implantation of "low-quality" embryos should be expanded to include mechanistic insights rather than relying solely on hypotheses. The inclusion of more examples that link spontaneous decidualization to specific diseases would further substantiate the findings. 

Discussion: The discussion should be enriched by addressing the potential for novel technologies, such as AI-driven image recognition, big data predictive analytics, organoids, and precision medicine, to assess endometrial receptivity. Exploring these emerging tools could provide valuable perspectives on future research directions in the field.

Comments on the Quality of English Language

None.

Author Response

Comment: The manuscript presents an intriguing idea, yet the content would benefit from a more detailed exploration of the underlying mechanisms, moving beyond general descriptions to focus on specific biological processes.

Response: We would like to thank the Reviewer for highlighting the strengths of our study as well as for pointing out several limitations. Below we tried to provide detailed explanation for each point. 

Comment 1: Lines 45-57: While it is widely recognized that decidualization varies across species, the manuscript should elaborate on these differences with greater specificity. For instance, it would be valuable to discuss variations in trophoblast types or unique gene manipulations that occur during this stage. Additionally, the manuscript should explore the cellular interactions involved and the signaling pathways through which the blastocyst communicates with the maternal endometrium. A concentrated figure or table summarizing these processes would enhance the reader's understanding.

Response1: Thanks for the suggestion. We substantially expanded this abstract. Firstly, we added information regarding the difference between establishment of fetal-maternal interface in marsupials and eutherians. Next, we described how decidual cells that evolved in eutherians enabled the endometrium to maintain the embryo for a prolonged period. In addition, we detailed the description of main types of eutherian placenta. We believe that the data presented in Figure 1 together with the updated description would enhance the reader's understanding. 

Previous version: “Decidualization of the endometrium appears alongside the development of invasive forms of the placenta in mammals, and the intensity of this reaction correlates with the depth of an embryo invasion into the tissue [13,14]. In cetaceans and ungulates with a non-invasive type of placenta, decidual cells are not detected [15]. In turn, in carnivores with an endotheliochorionic type of placenta, small isolated groups of decidual cells are found [14]. Finally, the most pronounced decidual reaction is observed in animals with a hemochorial type of placenta [14]. Notably, in the vast majority of animals characterized by decidualization, this reaction develops directly at the site of embryo implantation in response to the interaction of trophoblast cells with endometrial epithelial cells [16,17]. In other words, the decidual response of the maternal endometrium is controlled by signals emanating from a blastocyst.”

Updated version: “Even though mammalian viviparity evolved in the stem lineage of therian mammals, marsupials and eutherians employ distinct reproductive strategies, with a key difference in the establishment of the fetal-maternal interface [13-16] (Figure 1). For most of the duration of pregnancy in marsupials, the embryo is enclosed within an eggshell, while the contact between the fetal membranes and the uterine luminal epithelium occurs towards the end of the pregnancy, quickly followed by parturition. In most marsupials, the embryonic portion of the placenta is derived from the yolk sac, which may come into direct contact with the maternal endometrium but does not in-vade it. In contrast, embryo attachment during eutherian pregnancy leads to the estab-lishment of a stable fetal-maternal interface, which is essential for the formation of a definitive chorioallantoic placenta capable of sustaining prolonged pregnancy [13,14]. Intriguingly, in both marsupials and eutherians, fetal-maternal attachment is associat-ed with an inflammatory process; however, in marsupials, this results in parturition, whereas in eutherians, it leads to the establishment of a sustained fetal-maternal interface [15,16]. Current data suggest that this evolutionary transformation originates from a novel eutherian cell type – decidual stromal cells – which mediate the suppres-sion of the ancestral inflammatory response and enable the endometrium to maintain the embryo for a prolonged period. For instance, human decidual stromal cells have been recently shown to suppress IL17A production by T cells, thereby preventing in-filtration of neutrophils into the tissue [15]. Additional evidence regarding maternal adaptations to suppress uterine inflammation is described in the outstanding review [16]. Therefore, decidualization is a distinct feature of eutherian pregnancy, absent in marsupials (Figure 1).

The intensity of decidualization of the endometrium is associated with the depth of embryo invasion into the tissue [17,18]. Depending on the degree of endometrial invasion, three main types of eutherian placenta are recognized: hemochorial (the most invasive type, trophoblast cells penetrate the endometrial stroma and its vasculature to access maternal blood), endotheliochorial (a less invasive type, trophoblast interfaces with endometrial endothelial cells), and epitheliochorial (a non-invasive type, trophoblast interfaces with luminal epithelial cells of the endometrium) [19]. Expectedly, in cetaceans and ungulates with a non-invasive type of placenta, decidual cells are not detected [20]. In contrast, in carnivores with an endotheliochorial type of placenta, small isolated groups of decidual cells are found [18]. The most pronounced decidual reaction is observed in animals with a hemochorial type of placenta [18]. Notably, in the vast majority of animals characterized by decidualization, this reaction develops directly at the site of embryo implantation in response to the interaction between trophoblast cells and endometrial epithelial cells [16,21]. In other words, the decidual response of the maternal endometrium is controlled by signals emanating from a blastocyst”.

Comment 2: Lines 58-59: The relationship between hormonal regulation, menstruation, and decidualization should have a more detailed update. The manuscript needs to include recent findings from current studies, providing a comprehensive overview of how these factors interplay during the decidualization process.

Response 2: Thanks for the comment. We added suggested information into the text.

Previous version: “A different picture is typical for menstruating animals, including humans, in which decidualization occurs regardless of the presence of an embryo [17,18]. In this case, maternal hormones, primarily progesterone (P), are the only necessary inducers of decidualization. This type of decidualization induction is called “spontaneous”, since it does not depend on signals emanating from the embryo [19]. In humans, “spontaneous” decidualization begins with the luteal phase of the menstrual cycle within the first few days after ovulation and is driven by P produced by the corpus luteum appearing at the site of the ruptured dominant follicle. The development of “spontaneous” decidualization is thought to mediate the emergence of a new evolutionary feature - menstruation - the rejection of an unclaimed part of decidualized endometrial functionalis layer in response to a drop in P levels in cycles without conception [17]. Such an energy-consuming mechanism has evolved only in a few species of animals during evolution and is believed to provide them with certain advantages”.

Updated version: “A different picture is typical among menstruating animals, including humans, apes, elephant shrew, and certain bats [22-24]. For these species, decidualization is termed “spontaneous” because it occurs independently of contact with an embryo and is induced by maternal endocrine stimulation, specifically through the steroid hor-mone progesterone (P). According to recent findings, the responsiveness of endometrial stromal cells to P has evolved due to the continuous donation of the progesterone receptor (PR) binding sites to the regulatory regions of decidual genes by transposable elements in the eutherian lineage [25,26]. In humans, “spontaneous” decidualization begins during the luteal phase of the menstrual cycle, within the first few days fol-lowing ovulation, and is driven by P produced by the corpus luteum appearing at the site of the ruptured dominant follicle [16,27]. In a conception cycle, the implanting embryo secretes a significant amount of human chorionic gonadotrophin (hGC), an embryonic anti-luteolytic signal that sustains ovarian P production until the definitive placenta is formed. In non-conception cycle, the absence of embryonic hGC leads to the demise of the corpus luteum and a subsequent drop in the P levels, resulting in men-struation [16,27]. The initial phase of menstruation occurs in decidualized endometrial stromal cells, which sense P withdrawal due to PR expression [27]. Decidual stromal cells initiate a highly regulated inflammatory response that results in the production of various inflammatory cytokines, chemokines, and pro-inflammatory prostaglandins [28]. These inflammatory events recruit and activate immune cells, leading to extracel-lular matrix degradation and piecemeal tissue breakdown [29]. Therefore, the devel-opment of “spontaneous” decidualization is thought to mediate the emergence of a novel evolutionary feature – menstruation – the rejection of an unclaimed part of decidualized endometrial functionalis layer in response to a drop in P levels during cycles without conception [16]. Such an energy-consuming mechanism has evolved in only a few animal species during evolution and is believed to provide them with certain advantages."

Comment 3: Lines 71-84: The manuscript should delve deeper into the genetic changes that drive "spontaneous" decidualization, specifying the extent of these alterations. Additionally, the discussion around recurrent miscarriages and their role in restricting the implantation of "low-quality" embryos should be expanded to include mechanistic insights rather than relying solely on hypotheses. The inclusion of more examples that link spontaneous decidualization to specific diseases would further substantiate the findings.

Response 3: Thanks for the comment. We modified this paragraph as suggested.

Previous version: “According to modern concepts, emergence of “spontaneous” decidualization has mediated the shift of the control over the decidual reaction from embryonic to maternal one and thus underlined the development of the “sensory” ability of the maternal endometrium to an embryo [13]. This point of view finds a number of experimental confirmations. For example, in vitro models of implantation revealed that decidualizing human endometrial stromal cells (EnSC) exhibit a robust response against damaged embryos [19,20]. Specifically, the composition of factors secreted by decidual cells changes significantly in the presence of genetically abnormal embryos [20]. The authors of this study conclude that “spontaneous” decidualization developed during evolution to counterbalance the high incidence of chromosomal abnormalities in human embryos and enabled the maternal organism to restrict the implantation of “low-quality” embryos. This hypothesis was further supported by the findings that the endometrium of women with an impaired decidual reaction has reduced sensory function in the endometrium, leading to increased implantation rates but recurrent miscarriages [20,21]”.

Updated version: “According to modern concepts, the emergence of “spontaneous” decidualization has mediated the shift of the control of the decidual reaction from the embryonic to the maternal one and thus underlined the development of the “sensory” ability of the maternal endometrium to an embryo [17]. This point of view finds a number of experimental confirmations. For instance, in vitro models of implantation revealed that decidualizing human endometrial stromal cells (EnSC) exhibit a robust response to damaged embryos [23,24]. Specifically, the composition of factors secreted by decidual cells changes significantly in the presence of genetically abnormal embryos [24]. The authors of this study conclude that “spontaneous” decidualization evolved to counter-balance the high incidence of chromosomal abnormalities in human embryos and enabled the maternal organism to restrict the implantation of “low-quality” embryos. This hypothesis was further supported by the findings indicating that the endometrium of women with an impaired decidual reaction exhibits reduced sensory function, which leads to increased implantation rates but recurrent miscarriages [24, 30, 31]. Indeed, recurrent pregnancy loss has been associated with stem cell deficiency, accelerated stromal senescence, and deficient decidualization that limit the differentiating capacity of the endometrium and predispose individuals to pregnancy failure [31]. Additional evidence links defective functioning of EnSC and impaired decidualization to various complications during pregnancy [32-34]. For example, both endometriosis and adeno-myosis have been noted to express aggressive endometrial stem cells that display greater invasiveness [33]. Another pathology associated with defective decidualization is placenta accreta, a condition characterized by excessive invasion of the placenta into or through the uterine wall [34]. Since decidualization serves as a defense mechanism that protects the maternal organism from excessive embryo invasion, defects in this process can permit excessive placental invasion [32].”

Comment 4: Discussion: The discussion should be enriched by addressing the potential for novel technologies, such as AI-driven image recognition, big data predictive analytics, organoids, and precision medicine, to assess endometrial receptivity. Exploring these emerging tools could provide valuable perspectives on future research directions in the field.

Response4: Thanks for the suggestion. The information on artificial intelligence application was added into the appropriate part of the discussion.

Previous version: “The main methods used to assess the functional state of the endometrium include an ultrasound imaging of the pelvic organs and a pipelle biopsy of the tissue. These methods examines structure and thickness of the endometrium, as well as specific molecular markers (mostly, expression of steroid hormone receptors) using immuno-histochemical staining. Unfortunately, to date there is no compelling evidence of a significant correlation between the changes observed at the morphological / histological levels of the endometrium and the success of implantation in either natural or stimulated cycles [39–41]”.

Updated version: “The traditional methods used to assess the functional state of the endometrium include ultrasound imaging of the pelvic organs and pipelle biopsy of the tissue [53,54]. These methods examines structure and thickness of the endometrium, as well as specific molecular markers, primarily the expression of steroid hormone receptors, using immunohistochemical staining. Both approaches are highly operator-dependent and are therefore susceptible to significant biases arising from intra- and interobserver variations. To standardize interpretation and to enhance the reproducibility of these characteristics, more sophisticated approaches based on artificial intelligence (AI) are being developed [55-59]. For example, a recent study described a novel tool called EndoClassify, which utilizes AI to analyze ultrasonographic imaging to select optimal endometrial development prior to embryo transfer [56]. Another study exploits ultrasound images of the endometrium to train and test several AI models, with implantation as a binary outcome [57]. Furthermore, AI models are applied to identify CD138+ plasma cells within endometrial tissue and to assess endometrial histology features by calculating the areas occupied by epithelial and stromal cells [58,59]. Therefore, AI has the potential to improve the objectivity of ultrasonographic and histological examinations of the endometrium for both research and clinical purposes. At the same time, to date there is no compelling evidence of a significant correlation between the changes observed at the morphological / histological levels of the endometrium and the success of implantation in either natural or stimulated cycles [53,54,60].”.

Reviewer 3 Report

Comments and Suggestions for Authors

A typical review article like many in the literature. Written correctly. You can find much better publications in the literature

Author Response

Comment: A typical review article like many in the literature. Written correctly. You can find much better publications in the literature.

Response: We would like to thank the respected Reviewer for the time and effort spent to review our manuscript. We agree that the topic of the review is widely discussed in the current literature. However, we provide new ideas on how to use current knowledge on the decidualization process to develop more sophisticated instrument to assess receptivity. Particularly, information on evolutionary evolvement of decidualization along with the description of cellular events mediating this reaction, provided in the first chapters of the review, highlight the importance of stromal component, thus suggest to focus on gene signatures of stromal cells during decidualization rather than on whole endometrial tissue (as performed in the current tools). Additionally, we suggest unifying menstrual cycle length when developing new tool. Finally, we made substantial improvements of the first chapter in accordance with the comments of the second reviewer. Therefore, we believe that the present review is valuable for the field.